# Long-Term Analyses of the Rate of Perceived Exertion as an Indicator of Intensity in Women’s Basketball during a Relegation Play-off

**DOI:** 10.3390/biology11111592

**Published:** 2022-10-30

**Authors:** Abraham Batalla-Gavalda, Jose Vicente Beltran-Garrido, Gerson Garrosa-Martín, Pau Cecilia-Gallego, Raul Montoliu-Colás, Francisco Corbi

**Affiliations:** 1Health and Sport Sciences University School (EUSES), Rovira i Virgili University, 43870 Amposta, Spain; 2Sports Sciences Research Group INEFC Barcelona, Institut Nacional d’Educació Física de Catalunya (INEFC), Universitat de Barcelona, 08038 Barcelona, Spain; 3Department of Education and Specific Didactics, Faculty of Humanities and Social Sciences, Universitat Jaume I, 12071 Castellón de la Plana, Spain; 4Institute of New Imaging Technologies (INIT), Jaume I University, Avenue Vicente Sos Baynat, 12071 Castellón de la Plana, Spain; 5Institut Nacional d’Educación Física de Cataluña, Centre de Lleida, Universitat de Lleida, Complejo de La Caparrella, 25192 Lleida, Spain

**Keywords:** rate of perceived exertion, heart rate, basketball, relegation phase, women’s sport, amateur

## Abstract

**Simple Summary:**

Knowledge of internal load is essential to understand the effect of training and competition on athletes. The aim of this study was to analyse the rate of perceived exertion (RPE) scale as an indicator of intensity in amateur basketball players during a relegation play-off. The heart rate and RPE of 10 players from a Copa Catalunya team while competing over a 10-day period were analysed. There was an improvement in the relationship between the two variables from the first to the last match. The results suggest that RPE could be seen as an indicator of intensity in amateur basketball players during a relegation play-off, improving their relationship with MHR as the weeks went by, which could suggest a learning process.

**Abstract:**

Knowledge of internal load is essential to understand the effect of training and competition on athletes. The aim of this study was to analyse the validity of the rate of perceived exertion (RPE) scale as an indicator of intensity in amateur female basketball players during a relegation play-off. The heart rate and RPE of 10 players (age: 21.30 ± 2.71 years, weight: 68.84 ± 11.21 kg, body fat: 20.74 ± 3.51%) from a Copa Catalunya team while competing over a 10-day period were analysed. The mean heart rate of each match was registered with the Suunto Team Pack™ heart rate monitors. The RPE values were obtained once the match ended, completing the original Borg scale. The mean RPE ranged from 15.20 ± 2.39 to 18.00 ± 1.07 AU, whereas the mean heart rate (MHR) ranged from 132.35 ± 12.37 to 147.33 ± 10.61 bpm. There was also an improvement in the statistical correlation between the two variables as the days progressed. Regression equations were calculated for the total number of registered matches and the last five matches, obtaining the following regression equations: MHR = 6.23 × RPE20 + 36.8 (*R*^2^ = 0.56) for all games and MHR = 30.95 + 6.73 × RPE (*R*^2^ = 0.73) for the last five games. The results suggest that RPE could be seen as an indicator of intensity in amateur basketball players during a relegation play-off, improving their relationship with MHR as the weeks went by, which could suggest a learning process.

## 1. Introduction

There is some consensus on the benefits of sport and physical exercise on the physical and mental health of players, as well as on their psychological and social impact [1]. However, these will depend heavily on the ability to properly look at training loads on a case-by-case basis [2]. While it is true that a direct relationship has been established between the magnitude of the external load applied (duration and intensity), the physical fitness level, and the adaptations that the body is exposed to [3], several studies have described large differences between the external physical load applied and the ensuing internal effect [4]. The individual psychobiological response of the body to the training or competition loads generated by each situation is known as the internal load [2,5]. All this supports the idea that methodologies should be used during training and competition that provide case-by-case knowledge about its effect on the body [6].

The key factor of the load is the intensity and, from a physiological point of view, the knowledge of it during the competition is fundamental. Therefore, several indicators have been used to look at it on a case-by-case basis: maximal oxygen consumption, maximum heart rate [7,8], and aerobic and anaerobic thresholds, to name a few [2,9], with high levels of internal variability observed during monitoring. However, these variables also do not allow any differentiation to be made between the physiological and psychological effect of different types of training used and the environment where they are applied on athletes.

Therefore, various tools have been proposed to assess the psycho-physical effect of physical exercise on athletes, where rate of perceived exertion (RPE) is one that is most used [10,11,12]. The RPE provides knowledge of the perception of an individual about the level of intensity or what is the same: “How much have I exerted myself during an exercise, training session, or competition relative to my 100% best?” [13]. Its application is therefore subject to the feelings and sports players’ interoceptive control capacity [14,15], making it a particularly interesting tool to analyse the session in its entirety and in a sensitive, fast, and non-invasive way [16], since the results will be available immediately when the match is over. This can be extremely useful in anticipating and preventing overtraining and injury [10].

In fact, a number of studies, both on individual and team sports, have observed high levels of correlation between RPE and different physiological markers traditionally used to quantify load, such as heart rate (HR) [17,18], lactate in capillary blood [19], and the different metabolic pathways for obtaining energy [14]. Furthermore, while recognising that psychological factors have a physiological basis, psychological changes are considered more consistent than physiological changes, as they are able to influence them considerably, changing internal load values, especially when situations are potentially stressful to players [20].

However, on many occasions, and even more so when it comes to amateur sport, coaches very often do not have the technological resources to analyse the physiological needs of athletes to individualise the training load (such as heart rate monitors to measure heart rate), both during competition and during training. Furthermore, regulations themselves in sports do not allow the use of technological resources during competition. Therefore, the introduction of simple assessment systems such as RPE can be an interesting tool as a way of coming close to estimating the load of training and/or competition, in order to allow the physiological effect of a game situation without the need for technological devices.

In addition to the above paragraph, the introduction of the RPE as a tool for analysing internal load cannot be applied arbitrarily, as this could be modified depending on the player’s level (amateur vs. professional), the type of competition (regular vs. play off), and a learning process of the players when using this quantification method. In addition, we have to take into account the context in which we find ourselves when we carry out longitudinal analysis of play-off stint matches, which is highly demanding due to both the physiological fatigue that usually builds up in athletes after a long season and the psychological stress caused by the responsibility they are landed with in this type of event. For this reason, it is logical to think that the high specificity of this type of situation causes specific responses in the psycho-physiological dimensions of the athlete. This situation could also be aggravated by amateur athletes since they have neither the experience nor the tools to be able to manage this type of situation in many cases. Therefore, it is necessary to carry out studies to confirm its validity, depending on the population group analysed and the context in which the game takes place [21]. Although previous similar studies [22,23] have analysed the use of RPE as an indicator of intensity, these studies have not included a whole sample of women players competing during a real competitive situation such as a relegation phase.

Accordingly, the hypothesis of our study is that as the days of competition progress, the players will improve their perception in relation to the MHR, being able to declare a more adjusted value of RPE, which will increase the correlation between the variables, generating a learning process, and therefore it can be indicated that the RPE is a good indicator of intensity in amateur basketball players during a relegation play-off. The aim of this study was therefore to analyse the validity of the RPE scale as an indicator of intensity in amateur female basketball players during a relegation play-off in the Copa Catalunya.

## 2. Materials and Methods

### 2.1. Experimental Approach to the Problem

Following a pre-study familiarisation period of 4 weeks (from after Christmas week to the first week of February), a 11-week learning process (from the first week of February to mid-May, end of the competition) was considered to analyse the usefulness of the RPE scale as an indicator of intensity in female basketball players. Both trainings and matches and feedback learning moments occurred during the relegation play-off in the Copa Catalunya were used in this process. The players were monitored during all matches with heart rate monitors (Suunto^®^ Team Pack, Vantaa, Finland) in order to achieve the MHR values. After the training and competition, players were asked to declare their RPE (how much did I exert myself compared to my 100% exertion level?) using the Borg scale from 6 to 20 (original scale) [21]. Once the RPE was declared, the players were informed of the MHR obtained during the exercise to facilitate the learning process and the utilisation of the RPE scale. A schematic of the study protocol is shown in Figure 1.

### 2.2. Participants

A convenience sample of 10 amateur female basketball players from the same team of the Copa Catalunya (Spain) voluntarily participated in the study. Potential participants were informed of the risks and benefits of the study before signing the informed consent form. Only the participants who met the inclusion criteria were included in the study. The inclusion criteria of the study were as follows: (1) age between 18 and 35 years, (2) BMI status between 18.5 and 25.0 kg/m^2^, (3) belonging to an amateur team competing in a relegation play-off, (4) training at least three 120 min session plus a competitive match per week, (5) previous competitive experience of at least 5 years, and (6) not suffering from any injury or health issue that could impede the development of the intervention. The sample characteristics are shown in Table 1. All participants conducted three workouts per week with a 48 h recovery between sessions. The duration of each session was approximately 2 h (6 h per week), and all participants followed the same training. The participants were assigned a numerical code with the intention of safeguarding their identity, and a protocol was established for the delivery and explanation of the results. None of the study participants received financial or in-kind rewards for their participation. This study was conducted taking into account the principles of the Declaration of Helsinki for research with human beings [24].

### 2.3. Procedure

#### 2.3.1. Pre-Study Familiarisation

Before the relegation phase, both heart rate and declared RPE were recorded for 4 weeks after the lead training with the intention of familiarising participants with the study protocol (no records were made during the competition) [10]. No feedback was given to the participants in order to avoid learning during this process. Once the familiarisation was completed, the 11-week study (10 games and 33 trainings) began.

#### 2.3.2. Obtaining HR during Training and Competition

A total of 10 Suunto Team Pack™ heart rate monitors were used to obtain the mean HR. The heart rate monitors were placed 10 min before the start of the warm-up (10’ before the start of the training and during 36’ before the start of the matches (26’ warm-up)). HR was continually monitored for 10’ with the intention of assessing post-stress recovery after the end of each recording session.

#### 2.3.3. Video Recording of Events

All sessions were recorded using two film cameras (JVC–GZ620SE, HDD, Hong Kong, China), each placed at a height, allowing at least half a track to be recorded without the need for repositioning. To synchronise the whole game time of each match with the heart rate data, the cameras were synchronised with the heat rate monitors using an acoustic and visual signal. The synchronisation was repeated at the beginning and end of each quarter, and the video analysis was performed second by second.

#### 2.3.4. Obtaining the Mean HR of the Match

Following the end of the sporting event, the data were downloaded to a PC, and the MHR was calculated using the Whole Game concept, which is defined as the time from the start to the end of the match, reckoning the time in real time accounting [25].

#### 2.3.5. Obtaining the RPE after the Match

Once the match ended, the players completed the Original Borg scale [21] (a scale of 15 points [6,7,8,9,10,11,12,13,14,15,16,17,18,19,20], where 6 is a very, very slight perception, and 20 is a very, very hard perception) in response to the question “How much have I exerted myself in relation to my 100% best”? The questionnaire was administered 30’ after each game and 10’ after each training [23]. In order to prevent participants’ responses being influenced by those of their peers, the registration was carried out individually in an area set up for this purpose, and the participants were prevented from commenting on the marked result.

No type of intense physical activity was performed during the 48 h prior to each game in order to avoid any alteration in the perception of intensity, as a result of the effect of accumulated fatigue [14].

#### 2.3.6. Learning Process

Once MHR and RPE were obtained, each participant was informed of the two values recorded in that session (i.e., “The RPE of 7 that you have stated corresponds to an MHR for the session of 145 bpm”) and was compared with those recorded in previous sessions, in order to stimulate learning.

#### 2.3.7. Obtaining of Atmospheric Data

The temperature, relative humidity, and wind speed values for the 10 matches from the competition were analysed using a portable weather station Krestyle K4500 (Krestyle^®^, Boothwyn, PA, USA) in order to check whether the atmospheric environment could be a factor modifying the parameters obtained from both HR and RPE.

### 2.4. Statistical Analysis

The G*Power3 program for Mac Version 3.1.9.6 was used to calculate the sample size. The minimum effect size, for the relationship between HR and RPE, to which the statistical model was sensitive (*r* = 0.33; moderate) was determined by sensitivity analysis. The statistical test used was the correlation: point biserial model. The sample size obtained (*n* = 68) was estimated for an alpha error of 0.05 and a power of 0.80.

The normal distribution of the data was verified by the Shapiro–Wilk test. The data that did not fit the normal distribution were transformed using a logarithmic function, before proceeding to their analysis [26].

Pearson correlation coefficients (*r*) were calculated to assess the relationship between RPE and HR for each match. The strength of the coefficients *r* was interpreted as follows: trivial (<0.10), small (0.10–0.29), moderate (0.30–0.49), high (0.50–0.69), very high (0.70–0.89), or virtually perfect (>0.90) [27].

The free cocor statistical package for the R programming language was used to evaluate the differences in the relations between the RPE and the HR for each game.

A simple linear regression was used to determine the prediction of the HR value on the basis of the RPE value. To check the assumptions of normality and homocedasticity, the residuals versus predicted values, the standardised residuals histogram, and the Q-Q plot were inspected. The Durbin–Watson test was performed to evaluate collinearity [28]. To investigate whether the variance obtained improved as matches progressed, two different regression equations were calculated. The first one was calculated, including the data of all the games of the relegation phase, and the second one was calculated, including only the last 5 matches of the relegation phase.

The significance level was established at α = 0.10 in all tests following recommendations of Hopkins, W. G. et al. [27]. The analyses above were performed with the JASP program for Mac (Version 0.16.1; JASP Team, University of Amsterdam, Netherlands (2022)).

## 3. Results

A total of 68 valid RPE and HR data were recorded for the different games registered. The temperature, humidity, and wind speed data recorded in the matches were 13.48 ± 3.94 °C, 66.42 ± 14.83%, and 1.78 ± 0.82 m/s, respectively. The values of RPE and HR recorded in the different games are shown in Table 2 and are represented in Figure 2. The RPE vs. HR relationships for each match are shown in Figure 3. Correlation coefficients ranged from high (*r* = 0.56) in the second match to virtually perfect (*r* = 0.96) in the seventh match. Statistically significant differences were observed between the correlation coefficients between match 1 and match 7 (*p* = 0.065).

Linear regressions executed to predict HR values on the basis of RPE values are shown in Figure 4. The variable RPE (β1 = 0.69, *p* < 0.001) predicted the HR score of all registered matches, accounting for 56% of its variability. No collinearity was observed on the Durbin–Watson test (*D* = 2.33, *p* = 0.164). The heart rate equation obtained was HR = 36.83 + 6.23 × RPE (*p* < 0.001; *R*^2^ = 0.56) (see Figure 4A).

The variable RPE (β1 = 0.85, *p* < 0.001) predicted the HR score of the last five matches registered, accounting for 73% of its variability. No collinearity was observed on the Durbin–Watson test (*D* = 2.10, *p* = 0.767). The equation detail was HR = 30.95 + 6.73 × RPE (*p* < 0.001; *R*^2^ = 0.73) (see Figure 4B).

## 4. Discussion

This is the first study as far, as the authors are aware, in which the use of the RPE as a tool for controlling the intensity, both in official competition matches and in training in amateur women basketball players, is proposed using the original rate of perceived exertion scale [21].

The main finding of this work is the apparent correlation between MHR and RPE in amateur women basketball players, which improves as the games progress, indicating a possible learning process of at least 11 weeks to achieve a very high levels of correlation.

Prior to our study, the perception of effort in basketball was valued with different types of scales such as modified scales from 0 to 10 points or scales with pictograms, in reduced play situations or in base categories, and in professional players during competition and training. In this regard, Sampaio et al. [22] analysed a sample of male youth players in 3vs3 and 4vs4, concluding that the greater the number of players on the track, the higher the RPE. In contrast, Klusemann et al. [29] studied small-scale play situations involving 2vs2 and 4vs4 with 16 elite players in the junior category, indicating that the greater the number of players, the more slightly decreased the RPE. These differences could have been due to the inequality in the ages and genders of the participants, the size of the playing area (half-track or whole track), or the existence of differences in the density of work [11].

During competition, Calleja-González et al. [11], analysed a total of 150 basketball players of both genders in the youngster category using the scale with pictograms (from 0 to 10) developed by Parfitt et al. [30], obtaining mean RPE values of 4.48 ± 1.65 (mean ± SD). These authors also found that female players showed a greater perception of exertion. These higher values in the female gender appear to be maintained over time, since Piedra et al. [31] compared two teams of professional athletes of different gender over 141 sessions and obtained higher values for the female team (4.8 ± 1.52 vs. 4.24 ± 2.23 in men).

Although the comparison between different scales is complex, we consider the RPE values obtained in our study to be higher than those shown by other studies, since the lowest value obtained in our study was 15.2 ± 2.39. This value is equivalent to a perception greater than 7 on the 0-to-10-point scale. This difference could be determined by various factors such as the physical level of the players, the dynamics of the game itself, the defensive system applied [32], the number of rotations made during the matches, or the accumulated fatigue levels, both due to the normal development of the season and because of participants’ work obligations due to being amateurs. Although our study was conducted during a relegation phase, which could increase stress levels in players by altering their RPE, we cannot guarantee that the two are related due to not having data to compare relegation phase matches with regular matches. This aspect could be considered as a limitation of our study.

It must be highlighted that although the scales of 0 to 10 have been previously validated and correlated with HR, showing high (between *r* = 0.60 and *r* = 0.70) and very high correlations (between *r* = 0.83 and *r* = 0.85) [33]. The original Borg scale [21] was used in our study due to the fact that the level of correlation observed between the HR and each of the values on the scale is not only very high, but is higher than those obtained on the previous scales (from *r* = 0.80 to *r* = 0.90) [21].

The values obtained were between 132.35 ± 12.37 bpm and 147.33 ± 10.61 bpm in relation to the Whole Game MHR. These values were lower than those obtained by Vaquera Jiménez [34] in a sample composed of professional male players, who obtained different values on the basis of the position of the game: 163 ± 14.3 bpm for playmakers, 151 ± 10.3 bpm for forwards, and 155 ± 9.4 bpm for pivots. These differences could be due to the fact that the latter were men and from a competitive level, whereas the sample used in our study consisted of women and from a lower level. Various studies suggest that the higher the competitive level, the higher the recorded heart rate usually is [35]. Furthermore, the level of play also seems to have an influence, since higher level players seem to register lower heart rate levels [36]. Despite these results, it is worth mentioning that heart rate is usually indirectly related to play intensity [37], and its multifactorial component character makes it influenced by other factors such as play position, weather conditions under which play takes place, level of hydration, and eating habits [38].

Following analysis of the relationship between mean HR and RPE, it was observed that they seem to have a tendency to increase as the matches progressed, as values of *r* = 0.59 were obtained in the first match and *r* = 0.89. in the last. The mean value of the 10 matches analysed was *r* = 0.86, which could be considered a very high correlation [27]. These values are similar to those presented by Manzi et al. [39] in professional basketball players and those of Alexiou and Coutts [33], showing an *r* in soccer players ranging from 0.83 to 0.85.

The fact that the levels of correlation obtained in our work seem to improve as the matches advance could suggest the existence of a learning process. This trend made us choose to calculate two regression lines, with the intention of investigating whether the variance obtained improved as the matches progressed. To this end, the first was calculated including the data of all the games registered during the relegation phase (MHR = 6.23 × RPE20 + 36.8) and allowed for the explanation of a 56% variance of the results. A second regression line was then calculated, which included only the last five matches (MHR = 6.73 × RPE + 30.95), with the observation that this allowed for the explanation of a 73% variance of the results. This improvement in variance as the matches moved forward could suggest a learning process, improving as the matches progressed. The results obtained may suggest the use of RPE in population groups with little experience and in complex situations, such as a relegation phase, as long as a process of familiarisation and pre-learning is applied as a way of monitoring the internal load. This opens up the possibility of using RPE as an alternative low-cost and easy applicable tool, especially in equipment with lower resources.

This methodology would also allow for the monitoring of the internal responses derived from the external loads applied on each player during the competition, which is an economical alternative to the use of heart rate monitors that are normally non-authorised during competition.

## 5. Limitations

Despite the applicability of the methodology used in the present study, several considerations should be considered: firstly, that the regression lines obtained have been calculated in a sample of female amateur players and describe a specific competitive situation such as a relegation phase. This suggests the need to adapt this type of regression line to the competitive conditions of each team, since the situations could vary depending on the population group analysed, the type of competition, and the sport under study.

Secondly, the values obtained in the second regression line appear to be the result of a learning process. The duration of this process may be different depending on the features of the sample analysed and the contextual situation in which they are applied. Although 10 weeks of competition were used in this study, which would be equivalent to 11 calendar weeks (10 matches and 33 training sessions), there is a need for future studies in order to establish what its approximate duration should be. In addition, other factors that may modify the perception of fatigue, such as the menstrual cycle or the level of hydration, should be considered.

Despite these limitations, the use of regression lines allows for the use of heart rate monitors be disposed of or limited, reducing economic costs (each player must have their own unit), optimising time (saving a large amount of time by not having to download files for further processing and analysis), improving player convenience (the use of HR record bands can be uncomfortable and annoying for certain players), and preventing the loss of registers as a result of contact between players. They also allow for the interoceptive capacity of the players to be improved, who are forced to analyse their level of exertion from training to training, or game to game [33], thus improving their level of self-perception. In this regard, Gómez-Díaz et al. [14] suggest that the practical application of this method could become a fundamental tool for the coach or trainer, as it would guide the weekly training load while also monitoring the performance of the players in the match.

## 6. Conclusions

In conclusion, we can accept the hypothesis put forward in this study, and (1) it could be suggest that the use of RPE as an indicator of intensity is appropriate during a relegation phase in women’s basketball, due to the high correlations obtained with HR, but (2) before the RPE scale could be used as an indicator of intensity, a learning process must be carried out by the players. Therefore, subsequent studies that want to use the RPE as an indicator should take into account that not only the descriptive data has a value, but a learning process of at least 11 weeks should be carried out to improve the validity of the data.

## Figures and Tables

**Figure 1 biology-11-01592-f001:**
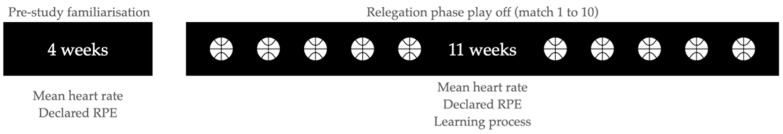
Schematic of the study protocol.

**Figure 2 biology-11-01592-f002:**
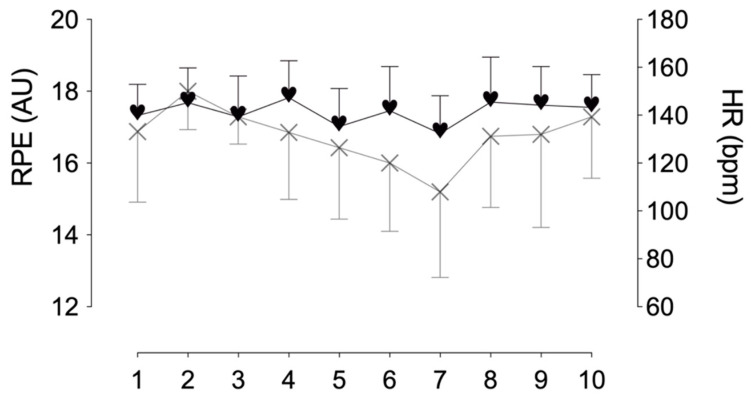
Mean values of RPE (rate of perceived exertion) and HR (mean heart rate) recorded in the different matches. Hearts represent the mean HR (bpm). Crosses represent RPE scores (arbitrary units). The bars indicate the standard deviation of the mean.

**Figure 3 biology-11-01592-f003:**
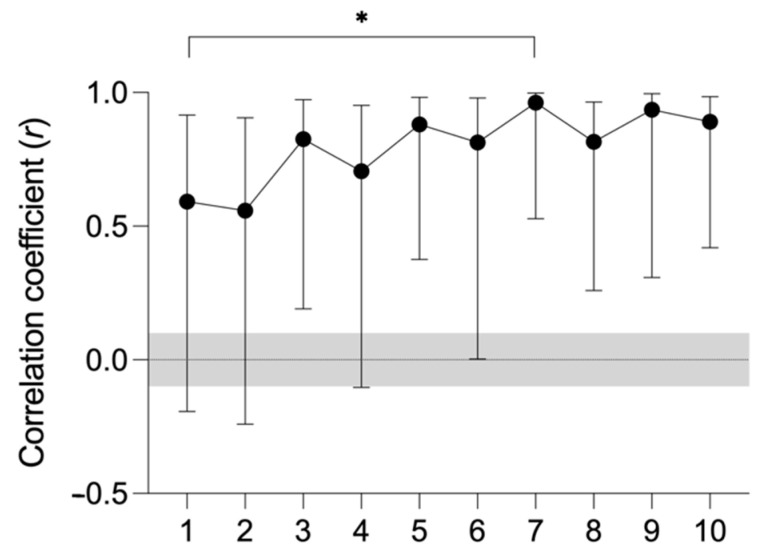
List of the values for RPE (rate of perceived exertion) and HR (mean heart rate) recorded in the different matches. The bars indicate uncertainty in correlation coefficients with 90% confidence intervals. The shaded area indicates the trivial effect size value. * *p* < 0.10: different from match 1.

**Figure 4 biology-11-01592-f004:**
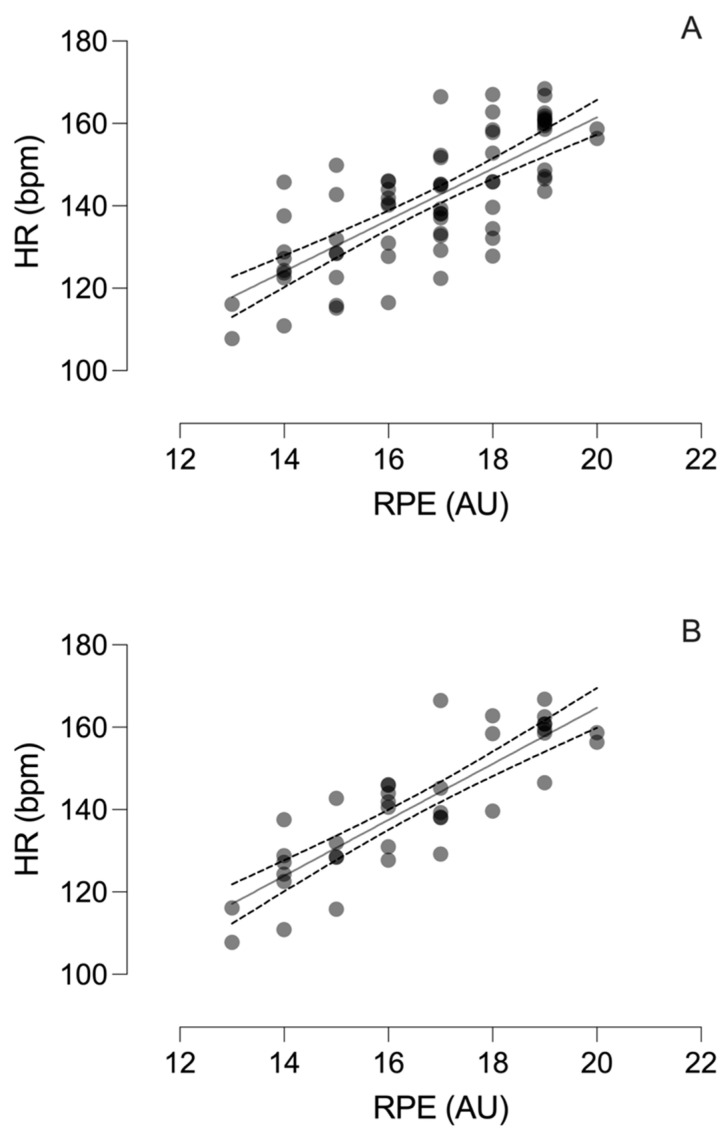
Linear relationship between the RPE (perceived stress scale) and HR (heart rate) variables. (**A**) Linear regression of all registered matches. HR = 36.83 + 6.23 × RPE (*p* < 0.001; *R*^2^ = 0.56). Dotted lines indicate the 90% confidence interval. (**B**) Linear regression of the last five matches registered. HR = 30.95 + 6.73 × RPE (*p* < 0.001; *R*^2^ = 0.73). Dotted lines indicate the 90% confidence interval.

**Table 1 biology-11-01592-t001:** Participants characteristics.

Age (Years)	Weight (kg)	Height (cm)	BMI (kg/m^2^)	Body Fat (%)	Competitive Experience (Years)
21.30 ± 2.71	68.84 ± 11.21	177.00 ± 7.00	21.73 ± 2.85	20.74 ± 3.51	10.00 ± 3.12

**Table 2 biology-11-01592-t002:** Means and standard deviations of the RPE reported by the players and the HR values corresponding to the whole game of each match.

Variable	M 1	M 2	M 3	M 4	M 5	M 6	M 7	M 8	M 9	M 10
RPE (AU)	16.88 ± 1.95	18.00 ± 1.07	17.29 ± 0.75	16.86 ± 1.86	16.43 ± 1.99	16.00 ± 1.90	15.20 ± 2.39	16.75 ± 1.98	16.80 ± 2.59	17.29 ± 1.70
HR (bpm)	139.90 ± 8.42	145.23 ± 9.41	139.37 ± 11.63	147.33 ± 10.61	135.21 ± 10.94	141.93 ± 13.46	132.35 ± 12.37	145.44 ± 12.22	144.22 ± 12.59	143.28 ± 9.35

The values are presented as mean ± SD. M: match; RPE: rate of perceived exertion, HR: heart rate; AU; arbitrary units; bpm: beats per minute.

## Data Availability

Data available on request due to restrictions, e.g., privacy or ethical restrictions.

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
