# Peer review of "Long-Term Analyses of the Rate of Perceived Exertion as an Indicator of Intensity in Women’s Basketball during a Relegation Play-off"

_biology, 2022, doi:10.3390/biology11111592_

Round 1
Reviewer 1 Report
The manuscript “Rate of perceived exertion as an indicator of intensity in women’s basketball” aimed to investigate the validity of using RPE as an indicator of exercise intensity during a relegation play-off in amateur female basketball players. I commend the authors for their work in conducting the experiment and preparing the manuscript. General comments and specific points and sections are provided below:
General comments
Presentation: overall, the presentation is good, with minor typos. The English writing is excellent and does require professional revisions. Tables and figures are high-quality.
Novelty: the information is novel considering the long-term analyses and, especially, the addition of psychological stress imposed by the relegation play off.
Title: the title is good, but lacks information on two of the main investigated factors: i) long-term analyses (and possible learning effects); ii) the psychological stress associated with a relegation play off. I recommend that the authors include these factors in the title, which would not only improve clarity but also highlight the novelty of the study.
Abstract:
Introduction: the introduction successfully presents the evidence on the use of session-RPE. However, some of the paragraphs appear to be disconnect. Most importantly, the novelty of the study appears to be the monitoring of session-RPE during relegation matches, and the effectiveness of this tool in a continuous competitive process. Therefore, I recommend that the authors further develop the rationale for these points (which have been briefly discussed only at the end of the introduction). Although important, the rationale presented before can be revised and shortened in order to give space for these specific points. Also, the authors fail to present the studies hypothesis at the end of the introduction.
L52: I do not think using “I” as an abbreviation for intensity is a good idea. Especially in English. Perhaps “INT” would be less misleading.
L66-67: It has been frequently recommended that session-RPE be assessed 15-30 min post-match/training sessions to avoid contamination from final efforts, therefore considering the session as a whole.
L70: Lactate
L78: Please do cite the first author’s name plus “et al.” before 22.
Materials and Method: this section is well structures and provides some information that was not provided in the introduction (e.g., match filming was not mentioned before). It is not clear to me, after reading the results, why the matches were filmed. One point that is also not clear is if there were inclusion criteria for the sample. I reckon that players who played only a few minutes would have a different perception of effort/HR response compared to key players who have played many minutes.
L140: continually?
L174-177: what dependent variable was used for sample size calculations?
L191: Is there a reason for the adoption of an alpha level of .10 instead of .05? If so, please include it in the description of the statistics.
Results: the results are clear and well presented, both graphically and textually. I commend the authors for their clarity and succinctness.
Discussion: the discussion is comprehensive and contextualizes the findings with the current literature. The writing is clear and possible limitations are presented. The interpretation of the data is sophisticated and I commend the authors for their excellent job in analyzing and discussing the data.
Conclusion: the conclusion is succinct and based-off the obtained results, as should be.
Reviewer 2 Report
Dear,
Manuscript Number: biology-1981267
Title Manuscript: Rate of Perceived Exertion as an Indicator of Intensity in Women's Basketball
This is a field method in the science of training for exercise physiology, as previous studies have reported rate of perceived exertion (RPE) as an indicator of exercise intensity in athletes and other people but at the moment MAJOR REVISIONS are necessary in order to make it suitable for a final decision.
POINTs of STRENGTH:
1) Assessment of rate of perceived exertion (RPE) as an indicator of exercise intensity using a field method in women's amateur basketball players.
POINTs of WEAKNESS (and/or should be revised to improve the manuscript):
Abstract:
2) The age, weight, BMI, and gender of participants are not specified in the methods section; please specify clearly;
3) Please provide the assessment of RPE and heart rate in the methods section of Abstract.
Keywords:
4) Please add keyword “Amateure“ in the keywords section.
Highlights
5) Please report the Highlights section based on the results obtained from the study.
Introduction:
6) The hypothesis and purpose of this study can be stated in more detail.
Materials and Methods:
7) The screening process of study participants and/ or exclusion and inclusion criteria should be described in more detail such as age, gender, BMI, physical level, pharmacological intervention and so on;
8) What is the novelty of your study? Please explain;
9) Was blood taken in this study? Please explain;
10) Did you evaluate other parameters as an indicator of fatigue or training intensity, such as lactic acid? If yes, please provide in the text of the manuscript.
11) Please provide a schematic of the study protocol in the manuscript.
Statistical analysis
12) If the authors used G-Power software to evaluate the sample size, add the type of statistical method used in the G-Power software;
13) The authors mentioned that they considered significant level at a p < 0.1? Was there a statistical reason? Please explain why the significance level of p < 0.05 was not considered?;
Discussion and Conclusions
14) What new information is sufficient to modify existing science of training?
15) What are the conclusions and implications for the science of training in this study? And particularly for future studies…?
16) What does this research add to the literature? Please clarify?
Limitations
17) As mentioned above, authors will agree that the limitation section has to be expanded.
References
18) References section is not always in accordance with the authors' guidelines. In particular, please check No. 1, 12, 17, 25, 26, 27, 31, 33, 34, 37, 40, 42, and 45 for validation. In addition, the mean age of the references in this manuscript is 2008 yrs; please Update.
Best Regards
Reviewer 3 Report
This study is interresting, but the way that was took to elaborate regression equations between the RPE and MHR is still not clear. Thus authors are requested to give more explanations regarding this point which is the most important in the study.
The statistical method is well designed
General comments.
Line 25 : add « of this study » after the aim
Ligne 30 : what do authors mean by « …..variables as the days progressed ». please consider modifying
Line 74 : IQ : must be explained as it first appears in the text
Line 109 : « The players were monitored at all times with pulse monitors.. ». I suppose during the game or training only ?
Line 113 : « Once the RPE was declared, the players were informed of the MHR obtained during the exercise and of the relationship between their RPE and HR. » on which basis the relationship was built and why this was declared to players ? Were they able to exploit it for next sessions ? furthermore does this not bias the results ?
Line 119 : Authors based their study on 10 basket ball players which i think is a small number, furthermore it is obvious how with those amateur basket ball players they trained only three times per week ? I think with 48h recovery between sessions they will get enough time to recover ; what do authors think about those points ?
Line 135 : Authors can add the period of the year in wich the familiarisation and the intervention was conducted ?
Did any relation was made between the intensity, the RPE and the menstrual cycle in participants ? this could be an interresting factor to explain the fatigue in some subjects.
Line 139 : « (10’ before the start of the training during and 36’ before the start of the matches » just change « and » before « during ».
Line 165 : « The RPE of 7 that you have stated corresponds to a MHR for the session of 145 bpm » How was this correspondance defined ?
Line 205 : According to figure 1 : how can authors explain this increase in RPE from match 7 to the match 10 ?
Line 234 : « The main finding of this work is the apparent correlation between MHR and RPE in 234 amateur basketball players, which improves as the games progress, indicating a possible 235 learning process. » my idea this progression can be more observed with beginners, but since the players have 10 years of experience , it is supposed that they have trained with high intensities ; the reviewer wonders if this study was conducted on the whole playing season what will be the results in that case ? Authors should take this point into consideration to discuss it in the discussion section.
Authors should discuss what was reported in the study of Sampaio et al. About the RPE score and the type of game performed in order to explain for the practitionner what could be the best method followed not to have high intensity, especially when beginners are training.
Line 295 : Authors must explain how these two regression equation were calculated ?
Line 299 : « This improvement in variance as the matches move forward could suggest a learning process, improving as the matches progressed. ». What is about if the opposite team does not afford intense effort during the game, can this for example bias the RPE responses. this should be discussed or evocated as a limitation point of the study.
Line 301 : this phrase should be reworded « Although this idea has not been proven in this study, the results obtained appear robust enough to consider the use of RPE in population groups with little experience and in complex situations, such as a relegation phase, as long as a process of familiarisation and pre-learning is applied as a way of monitoring the internal load. » since the idea has not been proven the results can not be robust till it is proven statistically.
Line 307-309 : « This methodology would also allow the monitoring of internal response (IL) that ex- ternal loads or stimuli produce in each player's body during competition, which is an alternative to the use of pulse monitors that are not normally authorised during their development. »
this is not clear enough and should be reworded. authors must be more accurate.
Line 336 : the conclusion is weak, the authors must stress more on the fact of the individualision of effort during the training sessions.
Round 2
Reviewer 2 Report
Dear,
Manuscript Number: biology-1981267
Title Manuscript: Long-Term Analyses of the Rate of Perceived Exertion as an Indicator of Intensity in Women’s Basketball during a Relegation Play -Off
Thank you very much for the efforts of the authors.
In general, this manuscript has found good content after correcting major revisions. Although revisions are modified and added in this manuscript, several concerns and/or REVISIONS have to be addressed before a final version can be made:
POINTs of WEAKNESS (and/or should be revised to improve the manuscript):
Abstract:
1) Please provide the assessment of RPE and heart rate in the methods section of Abstract.
Materials and Methods:
2) The screening process of study participants and/ or exclusion and inclusion criteria should be described in more detail;
3) What is the novelty of your study? Please explain;
4) Was blood taken in this study? Please explain;
5) Did you evaluate other parameters as an indicator of fatigue or training intensity, such as serum lactate test? If yes, please provide in the text of the manuscript.
Discussion and Conclusions
6) What new information is sufficient to modify the existing science of training?
7) What are the conclusions and implications for the science of training in this study? And particularly for future studies…?
8) What does this research add to the literature? Please clarify.
Limitations
9) As mentioned above, the authors will agree that the limitation section has to be expanded.
References
10) The mean age of the references in this manuscript is 2008 yrs; please Update.
Best Regards
21 October 2022
Round 3
Reviewer 2 Report
Dear Biology,
Manuscript Number: biology-1981267
Title Manuscript: Long-Term Analyses of the Rate of Perceived Exertion as an Indicator of Intensity in Women’s Basketball during a Relegation Play -Off
Thank you very much for the efforts of the authors. Although revisions are modified in this manuscript, several concerns/ MINOR REVISIONS have to be addressed before a final version can be made:
1) Based on previous similar studies, this sentence should be modified. “To the author’s knowledge, there is no study which analyze the use of RPE as an indicator of intensity in amateur women's basketball team during a relegation phase. This aspect is important due to the lack of information on women's sport in general, and specifically because of the specific situation of the relegation phase, a highly stressful competitive situation”;
2) The inclusion criteria of this study are not specified and should be carefully modified; inclusion criteria including age range, BMI status (normal, overweight, obese or all), physical fitness level or VO2max and/ or METs, what do the authors mean “amateur women basketball players”, based on this sentence “mean number of years competing of 10.00 ± 3.12 148 years”, Are athletes with 10 years of basketball experience considered amateur? Free of medication, alcohol, cigarettes, healthy status, and so on…
3) Please provide participant characteristics at baseline in the form of a Table … such as age, height, weight, BMI, VO2max, and other measured physiological characteristics of participants… ;
4) The authors in this manuscript reported that the sample size obtained (n = 68) was estimated for an alpha error of 0.05 and a power of 0.80. Were the total participants 10 basketball players and did authors evaluate the same 10 basketball players in ten matches? Please clearly specify;
4) What does this research add to the literature? Please clarify.
6) Please add a “limitations” section in this manuscript;
Best Regards
